# Automatic Variational Inference in Stan

**Alp Kucukelbir**
Columbia University
alp@cs.columbia.edu

**Rajesh Ranganath**
Princeton University
rajeshr@cs.princeton.edu

**Andrew Gelman**
Columbia University
gelman@stat.columbia.edu

**David M. Blei**
Columbia University
david.blei@columbia.edu

## Abstract

Variational inference is a scalable technique for approximate Bayesian inference. Deriving variational inference algorithms requires tedious model-specific calculations; this makes it difficult for non-experts to use. We propose an automatic variational inference algorithm, automatic differentiation variational inference (ADVI); we implement it in Stan (code available), a probabilistic programming system. In ADVI the user provides a Bayesian model and a dataset, nothing else. We make no conjugacy assumptions and support a broad class of models. The algorithm automatically determines an appropriate variational family and optimizes the variational objective. We compare ADVI to MCMC sampling across hierarchical generalized linear models, nonconjugate matrix factorization, and a mixture model. We train the mixture model on a quarter million images. With ADVI we can use variational inference on any model we write in Stan.

## 1 Introduction

Bayesian inference is a powerful framework for analyzing data. We design a model for data using latent variables; we then analyze data by calculating the posterior density of the latent variables. For machine learning models, calculating the posterior is often difficult; we resort to approximation.

Variational inference (VI) approximates the posterior with a simpler distribution [1, 2]. We search over a family of simple distributions and find the member closest to the posterior. This turns approximate inference into optimization. VI has had a tremendous impact on machine learning; it is typically faster than Markov chain Monte Carlo (MCMC) sampling (as we show here too) and has recently scaled up to massive data [3].

Unfortunately, VI algorithms are difficult to derive. We must first define the family of approximating distributions, and then calculate model-specific quantities relative to that family to solve the variational optimization problem. Both steps require expert knowledge. The resulting algorithm is tied to both the model and the chosen approximation.

In this paper we develop a method for automating variational inference, automatic differentiation variational inference (ADVI). Given any model from a wide class (specifically, probability models differentiable with respect to their latent variables), ADVI determines an appropriate variational family and an algorithm for optimizing the corresponding variational objective. We implement ADVI in Stan [4], a flexible probabilistic programming system. Stan describes a high-level language to define probabilistic models (e.g., Figure 2) as well as a model compiler, a library of transformations, and an efficient automatic differentiation toolbox. With ADVI we can now use variational inference on any model we write in Stan.[1] (See Appendices F to J.)

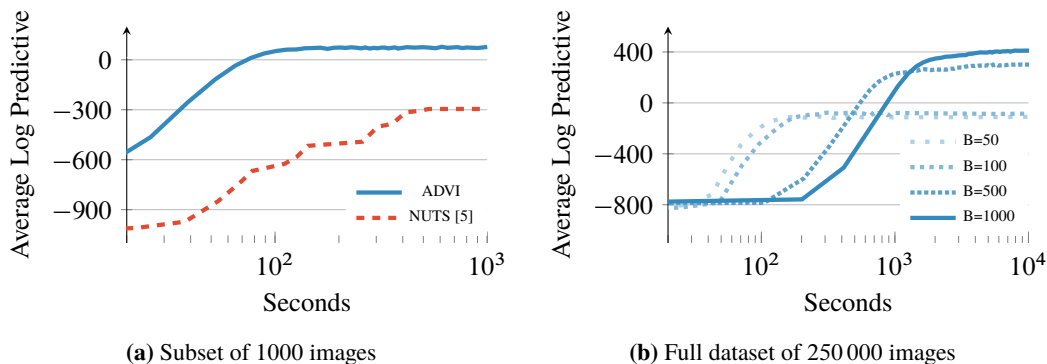

**(a)** Subset of 1000 images            **(b)** Full dataset of 250 000 images

**Figure 1:** Held-out predictive accuracy results | Gaussian mixture model (GMM) of the imageCLEF image histogram dataset. **(a)** ADVI outperforms the no-U-turn sampler (NUTS), the default sampling method in Stan [5]. **(b)** ADVI scales to large datasets by subsampling minibatches of size $B$ from the dataset at each iteration [3]. We present more details in Section 3.3 and Appendix J.

Figure 1 illustrates the advantages of our method. Consider a nonconjugate Gaussian mixture model for analyzing natural images; this is 40 lines in Stan (Figure 10). Figure 1a illustrates Bayesian inference on 1000 images. The $y$-axis is held-out likelihood, a measure of model fitness; the $x$-axis is time on a log scale. ADVI is orders of magnitude faster than NUTS, a state-of-the-art MCMC algorithm (and Stan's default inference technique) [5]. We also study nonconjugate factorization models and hierarchical generalized linear models in Section 3.

Figure 1b illustrates Bayesian inference on 250 000 images, the size of data we more commonly find in machine learning. Here we use ADVI with stochastic variational inference [3], giving an approximate posterior in under two hours. For data like these, MCMC techniques cannot complete the analysis.

**Related work.** ADVI automates variational inference within the Stan probabilistic programming system [4]. This draws on two major themes.

The first is a body of work that aims to generalize VI. Kingma and Welling [6] and Rezende et al. [7] describe a reparameterization of the variational problem that simplifies optimization. Ranganath et al. [8] and Salimans and Knowles [9] propose a black-box technique, one that only requires the model and the gradient of the approximating family. Titsias and Lázaro-Gredilla [10] leverage the gradient of the joint density for a small class of models. Here we build on and extend these ideas to automate variational inference; we highlight technical connections as we develop the method.

The second theme is probabilistic programming. Wingate and Weber [11] study VI in general probabilistic programs, as supported by languages like Church [12], Venture [13], and Anglican [14]. Another probabilistic programming system is infer.NET, which implements variational message passing [15], an efficient algorithm for conditionally conjugate graphical models. Stan supports a more comprehensive class of nonconjugate models with differentiable latent variables; see Section 2.1.

## 2  Automatic Differentiation Variational Inference

Automatic differentiation variational inference (ADVI) follows a straightforward recipe. First we transform the support of the latent variables to the real coordinate space. For example, the logarithm transforms a positive variable, such as a standard deviation, to the real line. Then we posit a Gaussian variational distribution to approximate the posterior. This induces a non-Gaussian approximation in the original variable space. Last we combine automatic differentiation with stochastic optimization to maximize the variational objective. We begin by defining the class of models we support.

### 2.1  Differentiable Probability Models

Consider a dataset $\mathbf{X} = x_{1:N}$ with $N$ observations. Each $x_n$ is a discrete or continuous random vector. The likelihood $p(\mathbf{X} \mid \boldsymbol{\theta})$ relates the observations to a set of latent random variables $\boldsymbol{\theta}$. Bayesian

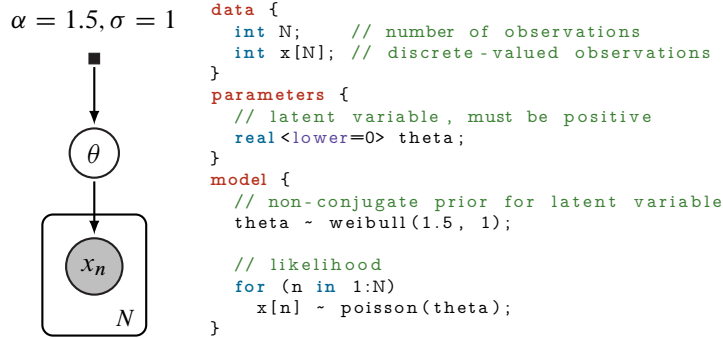

$\alpha = 1.5, \sigma = 1$

```
data {
    int N;      // number of observations
    int x[N];   // discrete-valued observations
}
parameters {
    // latent variable, must be positive
    real<lower=0> theta;
}
model {
    // non-conjugate prior for latent variable
    theta ~ weibull(1.5, 1);

    // likelihood
    for (n in 1:N)
        x[n] ~ poisson(theta);
}
```

**Figure 2:** Specifying a simple nonconjugate probability model in Stan.

analysis posits a prior density $p(\boldsymbol{\theta})$ on the latent variables. Combining the likelihood with the prior gives the joint density $p(\mathbf{X}, \boldsymbol{\theta}) = p(\mathbf{X} \mid \boldsymbol{\theta})\, p(\boldsymbol{\theta})$.

We focus on approximate inference for differentiable probability models. These models have continuous latent variables $\boldsymbol{\theta}$. They also have a gradient of the log-joint with respect to the latent variables $\nabla_{\boldsymbol{\theta}} \log p(\mathbf{X}, \boldsymbol{\theta})$. The gradient is valid within the support of the prior $\mathrm{supp}(p(\boldsymbol{\theta})) = \{ \boldsymbol{\theta} \mid \boldsymbol{\theta} \in \mathbb{R}^K \text{ and } p(\boldsymbol{\theta}) > 0 \} \subseteq \mathbb{R}^K$, where $K$ is the dimension of the latent variable space. This support set is important: it determines the support of the posterior density and plays a key role later in the paper. We make no assumptions about conjugacy, either full or conditional.[2]

For example, consider a model that contains a Poisson likelihood with unknown rate, $p(x \mid \theta)$. The observed variable $x$ is discrete; the latent rate $\theta$ is continuous and positive. Place a Weibull prior on $\theta$, defined over the positive real numbers. The resulting joint density describes a nonconjugate differentiable probability model. (See Figure 2.) Its partial derivative $\partial/\partial\theta\; p(x, \theta)$ is valid within the support of the Weibull distribution, $\mathrm{supp}(p(\theta)) = \mathbb{R}^+ \subset \mathbb{R}$. Because this model is nonconjugate, the posterior is not a Weibull distribution. This presents a challenge for classical variational inference. In Section 2.3, we will see how ADVI handles this model.

Many machine learning models are differentiable. For example: linear and logistic regression, matrix factorization with continuous or discrete measurements, linear dynamical systems, and Gaussian processes. Mixture models, hidden Markov models, and topic models have discrete random variables. Marginalizing out these discrete variables renders these models differentiable. (We show an example in Section 3.3.) However, marginalization is not tractable for all models, such as the Ising model, sigmoid belief networks, and (untruncated) Bayesian nonparametric models.

## 2.2  Variational Inference

Bayesian inference requires the posterior density $p(\boldsymbol{\theta} \mid \mathbf{X})$, which describes how the latent variables vary when conditioned on a set of observations $\mathbf{X}$. Many posterior densities are intractable because their normalization constants lack closed forms. Thus, we seek to approximate the posterior.

Consider an approximating density $q(\boldsymbol{\theta}\; ;\boldsymbol{\phi})$ parameterized by $\boldsymbol{\phi}$. We make no assumptions about its shape or support. We want to find the parameters of $q(\boldsymbol{\theta}\; ;\boldsymbol{\phi})$ to best match the posterior according to some loss function. Variational inference (VI) minimizes the Kullback-Leibler (KL) divergence from the approximation to the posterior [2],

$$\boldsymbol{\phi}^* = \arg\min_{\boldsymbol{\phi}} \mathrm{KL}(q(\boldsymbol{\theta}\; ;\boldsymbol{\phi}) \parallel p(\boldsymbol{\theta} \mid \mathbf{X})). \tag{1}$$

Typically the KL divergence also lacks a closed form. Instead we maximize the evidence lower bound (ELBO), a proxy to the KL divergence,

$$\mathcal{L}(\boldsymbol{\phi}) = \mathbb{E}_{q(\boldsymbol{\theta})}\big[ \log p(\mathbf{X}, \boldsymbol{\theta}) \big] - \mathbb{E}_{q(\boldsymbol{\theta})}\big[ \log q(\boldsymbol{\theta}\; ;\boldsymbol{\phi}) \big].$$

The first term is an expectation of the joint density under the approximation, and the second is the entropy of the variational density. Maximizing the ELBO minimizes the KL divergence [1, 16].

The minimization problem from Eq. (1) becomes

$$\boldsymbol{\phi}^* = \arg\max_{\boldsymbol{\phi}} \mathcal{L}(\boldsymbol{\phi}) \quad \text{such that} \quad \text{supp}(q(\boldsymbol{\theta}\,;\,\boldsymbol{\phi})) \subseteq \text{supp}(p(\boldsymbol{\theta}\mid\mathbf{X})). \tag{2}$$

We explicitly specify the support-matching constraint implied in the KL divergence.[3] We highlight this constraint, as we do not specify the form of the variational approximation; thus we must ensure that $q(\boldsymbol{\theta}\,;\,\boldsymbol{\phi})$ stays within the support of the posterior, which is defined by the support of the prior.

**Why is VI difficult to automate?** In classical variational inference, we typically design a conditionally conjugate model. Then the optimal approximating family matches the prior. This satisfies the support constraint by definition [16]. When we want to approximate models that are not conditionally conjugate, we carefully study the model and design custom approximations. These depend on the model and on the choice of the approximating density.

One way to automate VI is to use black-box variational inference [8, 9]. If we select a density whose support matches the posterior, then we can directly maximize the ELBO using Monte Carlo (MC) integration and stochastic optimization. Another strategy is to restrict the class of models and use a fixed variational approximation [10]. For instance, we may use a Gaussian density for inference in unrestrained differentiable probability models, i.e. where $\text{supp}(p(\boldsymbol{\theta})) = \mathbb{R}^K$.

We adopt a transformation-based approach. First we automatically transform the support of the latent variables in our model to the real coordinate space. Then we posit a Gaussian variational density. The transformation induces a non-Gaussian approximation in the original variable space and guarantees that it stays within the support of the posterior. Here is how it works.

## 2.3 Automatic Transformation of Constrained Variables

Begin by transforming the support of the latent variables $\boldsymbol{\theta}$ such that they live in the real coordinate space $\mathbb{R}^K$. Define a one-to-one differentiable function $T : \text{supp}(p(\boldsymbol{\theta})) \to \mathbb{R}^K$ and identify the transformed variables as $\boldsymbol{\zeta} = T(\boldsymbol{\theta})$. The transformed joint density $g(\mathbf{X}, \boldsymbol{\zeta})$ is

$$g(\mathbf{X}, \boldsymbol{\zeta}) = p\big(\mathbf{X}, T^{-1}(\boldsymbol{\zeta})\big)\big|\det J_{T^{-1}}(\boldsymbol{\zeta})\big|,$$

where $p$ is the joint density in the original latent variable space, and $J_{T^{-1}}$ is the Jacobian of the inverse of $T$. Transformations of continuous probability densities require a Jacobian; it accounts for how the transformation warps unit volumes [17]. (See Appendix D.)

Consider again our running example. The rate $\theta$ lives in $\mathbb{R}^+$. The logarithm $\zeta = T(\theta) = \log(\theta)$ transforms $\mathbb{R}^+$ to the real line $\mathbb{R}$. Its Jacobian adjustment is the derivative of the inverse of the logarithm, $|\det J_{T^{-1}(\zeta)}| = \exp(\zeta)$. The transformed density is

$$g(x, \zeta) = \text{Poisson}(x \mid \exp(\zeta))\,\text{Weibull}(\exp(\zeta)\,;\,1.5, 1)\,\exp(\zeta).$$

Figures 3a and 3b depict this transformation.

As we describe in the introduction, we implement our algorithm in Stan to enable generic inference. Stan implements a model compiler that automatically handles transformations. It works by applying a library of transformations and their corresponding Jacobians to the joint model density.[4] This transforms the joint density of any differentiable probability model to the real coordinate space. Now we can choose a variational distribution independent from the model.

## 2.4 Implicit Non-Gaussian Variational Approximation

After the transformation, the latent variables $\boldsymbol{\zeta}$ have support on $\mathbb{R}^K$. We posit a diagonal (mean-field) Gaussian variational approximation

$$q(\boldsymbol{\zeta}\,;\,\boldsymbol{\phi}) = \mathcal{N}(\boldsymbol{\zeta}\,;\,\boldsymbol{\mu}, \boldsymbol{\sigma}) = \prod_{k=1}^{K} \mathcal{N}(\zeta_k\,;\,\mu_k, \sigma_k).$$

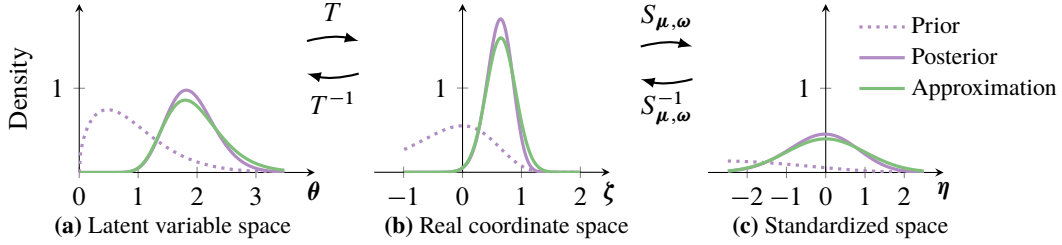

**Figure 3:** Transformations for ADVI. The purple line is the posterior. The green line is the approximation. **(a)** The latent variable space is $\mathbb{R}^+$. **(a→b)** $T$ transforms the latent variable space to $\mathbb{R}$. **(b)** The variational approximation is a Gaussian. **(b→c)** $S_{\boldsymbol{\mu},\boldsymbol{\omega}}$ absorbs the parameters of the Gaussian. **(c)** We maximize the ELBO in the standardized space, with a fixed standard Gaussian approximation.

The vector $\boldsymbol{\phi} = (\mu_1, \cdots, \mu_K, \sigma_1, \cdots, \sigma_K)$ contains the mean and standard deviation of each Gaussian factor. This defines our variational approximation in the real coordinate space. (Figure 3b.)

The transformation $T$ maps the support of the latent variables to the real coordinate space; its inverse $T^{-1}$ maps back to the support of the latent variables. This implicitly defines the variational approximation in the original latent variable space as $q(T(\boldsymbol{\theta})\,;\,\boldsymbol{\phi})\big|\det J_T(\boldsymbol{\theta})\big|$. The transformation ensures that the support of this approximation is always bounded by that of the true posterior in the original latent variable space (Figure 3a). Thus we can freely optimize the ELBO in the real coordinate space (Figure 3b) without worrying about the support matching constraint.

The ELBO in the real coordinate space is

$$\mathscr{L}(\boldsymbol{\mu},\boldsymbol{\sigma}) = \mathbb{E}_{q(\boldsymbol{\zeta})}\bigg[\log p\big(\mathbf{X}, T^{-1}(\boldsymbol{\zeta})\big) + \log\big|\det J_{T^{-1}}(\boldsymbol{\zeta})\big|\bigg] + \frac{K}{2}\left(1 + \log(2\pi)\right) + \sum_{k=1}^{K}\log\sigma_k,$$

where we plug in the analytic form of the Gaussian entropy. (The derivation is in Appendix A.)

We choose a diagonal Gaussian for efficiency. This choice may call to mind the Laplace approximation technique, where a second-order Taylor expansion around the maximum-a-posteriori estimate gives a Gaussian approximation to the posterior. However, using a Gaussian variational approximation is not equivalent to the Laplace approximation [18]. The Laplace approximation relies on maximizing the probability density; it fails with densities that have discontinuities on its boundary. The Gaussian approximation considers probability mass; it does not suffer this degeneracy. Furthermore, our approach is distinct in another way: because of the transformation, the posterior approximation in the original latent variable space (Figure 3a) is non-Gaussian.

## 2.5 Automatic Differentiation for Stochastic Optimization

We now maximize the ELBO in real coordinate space,

$$\boldsymbol{\mu}^*, \boldsymbol{\sigma}^* = \arg\max_{\boldsymbol{\mu},\boldsymbol{\sigma}} \mathscr{L}(\boldsymbol{\mu},\boldsymbol{\sigma}) \quad \text{such that} \quad \boldsymbol{\sigma} \succ 0. \tag{3}$$

We use gradient ascent to reach a local maximum of the ELBO. Unfortunately, we cannot apply automatic differentiation to the ELBO in this form. This is because the expectation defines an intractable integral that depends on $\boldsymbol{\mu}$ and $\boldsymbol{\sigma}$; we cannot directly represent it as a computer program. Moreover, the standard deviations in $\boldsymbol{\sigma}$ must remain positive. Thus, we employ one final transformation: elliptical standardization[5] [19], shown in Figures 3b and 3c.

First re-parameterize the Gaussian distribution with the log of the standard deviation, $\boldsymbol{\omega} = \log(\boldsymbol{\sigma})$, applied element-wise. The support of $\boldsymbol{\omega}$ is now the real coordinate space and $\boldsymbol{\sigma}$ is always positive. Then define the standardization $\boldsymbol{\eta} = S_{\boldsymbol{\mu},\boldsymbol{\omega}}(\boldsymbol{\zeta}) = \text{diag}\big(\exp(\boldsymbol{\omega})^{-1}\big)(\boldsymbol{\zeta} - \boldsymbol{\mu})$. The standardization

**Algorithm 1:** Automatic differentiation variational inference (ADVI)
---

    **Input**: Dataset $\mathbf{X} = \boldsymbol{x}_{1:N}$, model $p(\mathbf{X}, \boldsymbol{\theta})$.
    Set iteration counter $i = 0$ and choose a stepsize sequence $\boldsymbol{\rho}^{(i)}$.
    Initialize $\boldsymbol{\mu}^{(0)} = \mathbf{0}$ and $\boldsymbol{\omega}^{(0)} = \mathbf{0}$.

    **while** *change in* ELBO *is above some threshold* **do**

        Draw $M$ samples $\boldsymbol{\eta}_m \sim \mathcal{N}(\mathbf{0}, \mathbf{I})$ from the standard multivariate Gaussian.
        Invert the standardization $\boldsymbol{\zeta}_m = \mathrm{diag}(\exp{(\boldsymbol{\omega}^{(i)})})\boldsymbol{\eta}_m + \boldsymbol{\mu}^{(i)}$.

        Approximate $\nabla_{\boldsymbol{\mu}}\mathcal{L}$ and $\nabla_{\boldsymbol{\omega}}\mathcal{L}$ using MC integration (Eqs. (4) and (5)).

        Update $\boldsymbol{\mu}^{(i+1)} \longleftarrow \boldsymbol{\mu}^{(i)} + \boldsymbol{\rho}^{(i)}\nabla_{\boldsymbol{\mu}}\mathcal{L}$ and $\boldsymbol{\omega}^{(i+1)} \longleftarrow \boldsymbol{\omega}^{(i)} + \boldsymbol{\rho}^{(i)}\nabla_{\boldsymbol{\omega}}\mathcal{L}$.

        Increment iteration counter.

    **end**
    Return $\boldsymbol{\mu}^* \longleftarrow \boldsymbol{\mu}^{(i)}$ and $\boldsymbol{\omega}^* \longleftarrow \boldsymbol{\omega}^{(i)}$.
---

encapsulates the variational parameters and gives the fixed density

$$q(\boldsymbol{\eta}\,;\,\mathbf{0}, \mathbf{I}) = \mathcal{N}(\boldsymbol{\eta}\,;\,\mathbf{0}, \mathbf{I}) = \prod_{k=1}^{K} \mathcal{N}(\eta_k\,;\,0, 1).$$

The standardization transforms the variational problem from Eq. (3) into

$$
\begin{aligned}
\boldsymbol{\mu}^*, \boldsymbol{\omega}^* &= \underset{\boldsymbol{\mu}, \boldsymbol{\omega}}{\arg\max}\,\mathcal{L}(\boldsymbol{\mu}, \boldsymbol{\omega}) \\
&= \underset{\boldsymbol{\mu}, \boldsymbol{\omega}}{\arg\max}\,\mathbb{E}_{\mathcal{N}(\boldsymbol{\eta}\,;\,\mathbf{0},\mathbf{I})}\Bigg[ \log p\big(\mathbf{X}, T^{-1}(S_{\boldsymbol{\mu}, \boldsymbol{\omega}}^{-1}(\boldsymbol{\eta}))\big) + \log\big|\det J_{T^{-1}}\big(S_{\boldsymbol{\mu},\boldsymbol{\omega}}^{-1}(\boldsymbol{\eta})\big)\big| \Bigg] + \sum_{k=1}^{K}\omega_k,
\end{aligned}
$$

where we drop constant terms from the calculation. This expectation is with respect to a standard Gaussian and the parameters $\boldsymbol{\mu}$ and $\boldsymbol{\omega}$ are both unconstrained (Figure 3c). We push the gradient inside the expectations and apply the chain rule to get

$$\nabla_{\boldsymbol{\mu}}\mathcal{L} = \mathbb{E}_{\mathcal{N}(\boldsymbol{\eta})}\big[\nabla_{\boldsymbol{\theta}}\log p(\mathbf{X}, \boldsymbol{\theta})\nabla_{\boldsymbol{\zeta}}T^{-1}(\boldsymbol{\zeta}) + \nabla_{\boldsymbol{\zeta}}\log\big|\det J_{T^{-1}}(\boldsymbol{\zeta})\big|\big], \tag{4}$$

$$\nabla_{\omega_k}\mathcal{L} = \mathbb{E}_{\mathcal{N}(\eta_k)}\big[\big(\nabla_{\theta_k}\log p(\mathbf{X}, \boldsymbol{\theta})\nabla_{\zeta_k}T^{-1}(\boldsymbol{\zeta}) + \nabla_{\zeta_k}\log\big|\det J_{T^{-1}}(\boldsymbol{\zeta})\big|\big)\,\eta_k\exp(\omega_k)\big] + 1. \tag{5}$$

(The derivations are in Appendix B.)

We can now compute the gradients inside the expectation with automatic differentiation. The only thing left is the expectation. MC integration provides a simple approximation: draw $M$ samples from the standard Gaussian and evaluate the empirical mean of the gradients within the expectation [20].

This gives unbiased noisy gradients of the ELBO for any differentiable probability model. We can now use these gradients in a stochastic optimization routine to automate variational inference.

## 2.6 Automatic Variational Inference

Equipped with unbiased noisy gradients of the ELBO, ADVI implements stochastic gradient ascent (Algorithm 1). We ensure convergence by choosing a decreasing step-size sequence. In practice, we use an adaptive sequence [21] with finite memory. (See Appendix E for details.)

ADVI has complexity $\mathcal{O}(2NMK)$ per iteration, where $M$ is the number of MC samples (typically between 1 and 10). Coordinate ascent VI has complexity $\mathcal{O}(2NK)$ per pass over the dataset. We scale ADVI to large datasets using stochastic optimization [3, 10]. The adjustment to Algorithm 1 is simple: sample a minibatch of size $B \ll N$ from the dataset and scale the likelihood of the sampled minibatch by $N/B$ [3]. The stochastic extension of ADVI has per-iteration complexity $\mathcal{O}(2BMK)$.

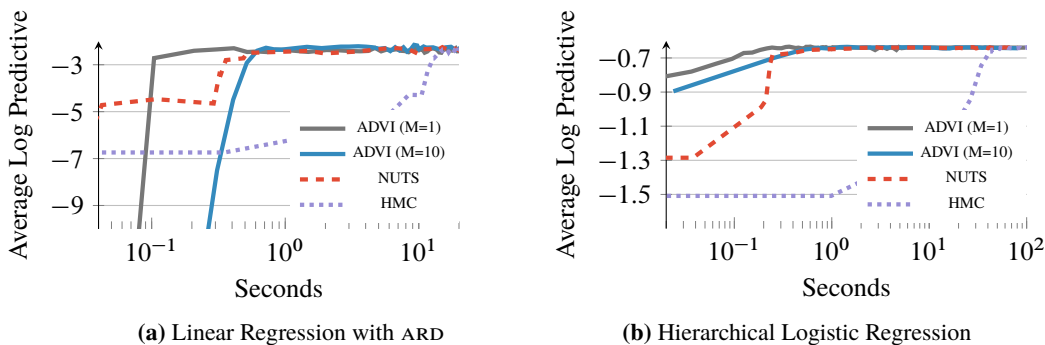

**(a)** Linear Regression with ARD  **(b)** Hierarchical Logistic Regression

**Figure 4:** Hierarchical generalized linear models. Comparison of ADVI to MCMC: held-out predictive likelihood as a function of wall time.

## 3 Empirical Study

We now study ADVI across a variety of models. We compare its speed and accuracy to two Markov chain Monte Carlo (MCMC) sampling algorithms: Hamiltonian Monte Carlo (HMC) [22] and the no-U-turn sampler (NUTS)[6] [5]. We assess ADVI convergence by tracking the ELBO. To place ADVI and MCMC on a common scale, we report predictive likelihood on held-out data as a function of time. We approximate the posterior predictive likelihood using a MC estimate. For MCMC, we plug in posterior samples. For ADVI, we draw samples from the posterior approximation during the optimization. We initialize ADVI with a draw from a standard Gaussian.

We explore two hierarchical regression models, two matrix factorization models, and a mixture model. All of these models have nonconjugate prior structures. We conclude by analyzing a dataset of 250 000 images, where we report results across a range of minibatch sizes $B$.

### 3.1 A Comparison to Sampling: Hierarchical Regression Models

We begin with two nonconjugate regression models: linear regression with automatic relevance determination (ARD) [16] and hierarchical logistic regression [23].

**Linear Regression with ARD.** This is a sparse linear regression model with a hierarchical prior structure. (Details in Appendix F.) We simulate a dataset with 250 regressors such that half of the regressors have no predictive power. We use 10 000 training samples and hold out 1000 for testing.

**Logistic Regression with Spatial Hierarchical Prior.** This is a hierarchical logistic regression model from political science. The prior captures dependencies, such as states and regions, in a polling dataset from the United States 1988 presidential election [23]. (Details in Appendix G.) We train using 10 000 data points and withhold 1536 for evaluation. The regressors contain age, education, state, and region indicators. The dimension of the regression problem is 145.

**Results.** Figure 4 plots average log predictive accuracy as a function of time. For these simple models, all methods reach the same predictive accuracy. We study ADVI with two settings of $M$, the number of MC samples used to estimate gradients. A single sample per iteration is sufficient; it is also the fastest. (We set $M = 1$ from here on.)

### 3.2 Exploring Nonconjugacy: Matrix Factorization Models

We continue by exploring two nonconjugate non-negative matrix factorization models: a constrained Gamma Poisson model [24] and a Dirichlet Exponential model. Here, we show how easy it is to explore new models using ADVI. In both models, we use the Frey Face dataset, which contains 1956 frames (28 × 20 pixels) of facial expressions extracted from a video sequence.

**Constrained Gamma Poisson.** This is a Gamma Poisson factorization model with an ordering constraint: each row of the Gamma matrix goes from small to large values. (Details in Appendix H.)

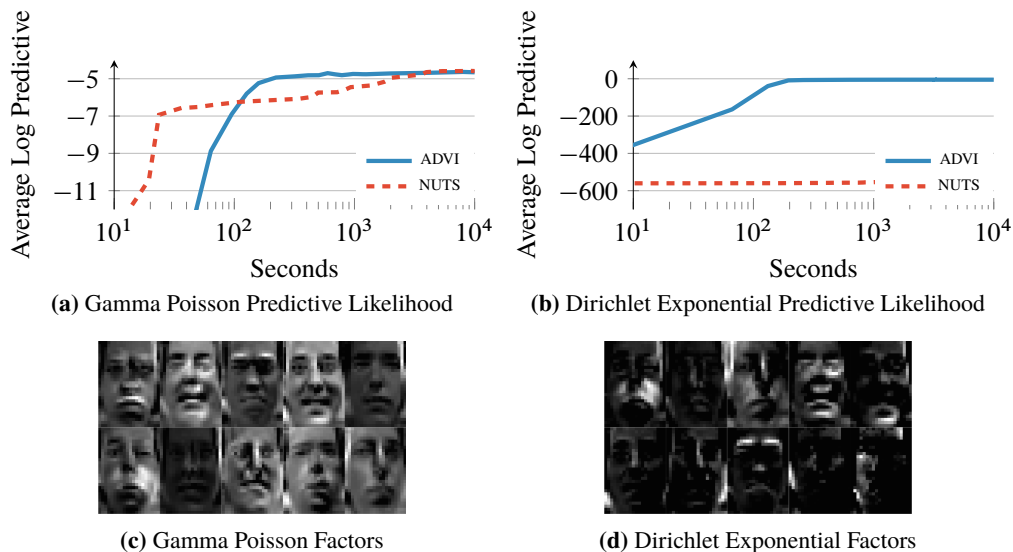

**(a)** Gamma Poisson Predictive Likelihood      **(b)** Dirichlet Exponential Predictive Likelihood

**(c)** Gamma Poisson Factors      **(d)** Dirichlet Exponential Factors

**Figure 5:** Non-negative matrix factorization of the Frey Faces dataset. Comparison of ADVI to MCMC: held-out predictive likelihood as a function of wall time.

**Dirichlet Exponential.** This is a nonconjugate Dirichlet Exponential factorization model with a Poisson likelihood. (Details in Appendix I.)

**Results.** Figure 5 shows average log predictive accuracy as well as ten factors recovered from both models. ADVI provides an order of magnitude speed improvement over NUTS (Figure 5a). NUTS struggles with the Dirichlet Exponential model (Figure 5b). In both cases, HMC does not produce any useful samples within a budget of one hour; we omit HMC from the plots.

### 3.3 Scaling to Large Datasets: Gaussian Mixture Model

We conclude with the Gaussian mixture model (GMM) example we highlighted earlier. This is a nonconjugate GMM applied to color image histograms. We place a Dirichlet prior on the mixture proportions, a Gaussian prior on the component means, and a lognormal prior on the standard deviations. (Details in Appendix J.) We explore the imageCLEF dataset, which has 250 000 images [25]. We withhold 10 000 images for evaluation.

In Figure 1a we randomly select 1000 images and train a model with 10 mixture components. NUTS struggles to find an adequate solution and HMC fails altogether. This is likely due to label switching, which can affect HMC-based techniques in mixture models [4].

Figure 1b shows ADVI results on the full dataset. Here we use ADVI with stochastic subsampling of minibatches from the dataset [3]. We increase the number of mixture components to 30. With a minibatch size of 500 or larger, ADVI reaches high predictive accuracy. Smaller minibatch sizes lead to suboptimal solutions, an effect also observed in [3]. ADVI converges in about two hours.

## 4 Conclusion

We develop automatic differentiation variational inference (ADVI) in Stan. ADVI leverages automatic transformations, an implicit non-Gaussian variational approximation, and automatic differentiation. This is a valuable tool. We can explore many models and analyze large datasets with ease. We emphasize that ADVI is currently available as part of Stan; it is ready for anyone to use.

**Acknowledgments**

We thank Dustin Tran, Bruno Jacobs, and the reviewers for their comments. This work is supported by NSF IIS-0745520, IIS-1247664, IIS-1009542, SES-1424962, ONR N00014-11-1-0651, DARPA FA8750-14-2-0009, N66001-15-C-4032, Sloan G-2015-13987, IES DE R305D140059, NDSEG, Facebook, Adobe, Amazon, and the Siebel Scholar and John Templeton Foundations.

## Footnotes

[1] ADVI is available in Stan 2.8. See Appendix C.

[2] The posterior of a *fully* conjugate model is in the same family as the prior; a *conditionally* conjugate model has this property within the complete conditionals of the model [3].

[3]If $\text{supp}(q) \not\subseteq \text{supp}(p)$ then outside the support of $p$ we have $\text{KL}(q \parallel p) = \mathbb{E}_q[\log q] - \mathbb{E}_q[\log p] = -\infty$.

[4]Stan provides transformations for upper and lower bounds, simplex and ordered vectors, and structured matrices such as covariance matrices and Cholesky factors [4].

[5]Also known as a "co-ordinate transformation" [7], an "invertible transformation" [10], and the "reparameterization trick" [6].

[6]NUTS is an adaptive extension of HMC. It is the default sampler in Stan.

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
