[Supplementary Material]

## A  Transformation of the Evidence Lower Bound

Recall that $\boldsymbol{\zeta} = T(\boldsymbol{\theta})$ and that the variational approximation in the real coordinate space is $q(\boldsymbol{\zeta}\,;\boldsymbol{\phi})$.

We begin with the evidence lower bound (ELBO) in the original latent variable space. We then transform the latent variable space of to the real coordinate space.

$$
\begin{aligned}
\mathcal{L} &= \int q(\boldsymbol{\theta}\,;\boldsymbol{\phi}) \log\left[\frac{p(\mathbf{X},\boldsymbol{\theta})}{q(\boldsymbol{\theta}\,;\boldsymbol{\phi})}\right] \mathrm{d}\boldsymbol{\theta} \\
&= \int q(\boldsymbol{\zeta}\,;\boldsymbol{\phi}) \log\left[\frac{p\big(\mathbf{X},T^{-1}(\boldsymbol{\zeta})\big)\big|\det J_{T^{-1}}(\boldsymbol{\zeta})\big|}{q(\boldsymbol{\zeta}\,;\boldsymbol{\phi})}\right] \mathrm{d}\boldsymbol{\zeta} \\
&= \int q(\boldsymbol{\zeta}\,;\boldsymbol{\phi}) \log\left[p\big(\mathbf{X},T^{-1}(\boldsymbol{\zeta})\big)\big|\det J_{T^{-1}}(\boldsymbol{\zeta})\big|\right] \mathrm{d}\boldsymbol{\zeta} - \int q(\boldsymbol{\zeta}\,;\boldsymbol{\phi}) \log\left[q(\boldsymbol{\zeta}\,;\boldsymbol{\phi})\right] \mathrm{d}\boldsymbol{\zeta} \\
&= \mathbb{E}_{q(\boldsymbol{\zeta})}\left[\log p\big(\mathbf{X},T^{-1}(\boldsymbol{\zeta})\big) + \log\big|\det J_{T^{-1}}(\boldsymbol{\zeta})\big|\right] - \mathbb{E}_{q(\boldsymbol{\zeta})}\left[\log q(\boldsymbol{\zeta}\,;\boldsymbol{\phi})\right]
\end{aligned}
$$

The variational approximation in the real coordinate space is a Gaussian. Plugging in its entropy gives the ELBO in the real coordinate space

$$
\mathcal{L} = \mathbb{E}_{q(\boldsymbol{\zeta})}\left[\log p\big(\mathbf{X},T^{-1}(\boldsymbol{\zeta})\big) + \log\big|\det J_{T^{-1}}(\boldsymbol{\zeta})\big|\right] + \frac{1}{2}K\,(1+\log(2\pi)) + \sum_{k=1}^{K}\log\sigma_k.
$$

## B  Gradients of the Evidence Lower Bound

First, consider the gradient with respect to the $\boldsymbol{\mu}$ parameter of the standardization. We exchange the order of the gradient and the integration through the dominated convergence theorem [1]. The rest is the chain rule for differentiation.

$$
\begin{aligned}
\nabla_{\boldsymbol{\mu}}\mathcal{L} &= \nabla_{\boldsymbol{\mu}}\Bigg\{\mathbb{E}_{\mathcal{N}(\boldsymbol{\eta}\,;\mathbf{0},\mathbf{I})}\left[\log p\big(\mathbf{X},T^{-1}(S_{\boldsymbol{\mu},\boldsymbol{\omega}}^{-1}(\boldsymbol{\eta}))\big) + \log\big|\det J_{T^{-1}}\big(S_{\boldsymbol{\mu},\boldsymbol{\omega}}^{-1}(\boldsymbol{\eta})\big)\big|\right] \\
&\qquad + \frac{K}{2}(1+\log(2\pi)) + \sum_{k=1}^{K}\log\sigma_k\Bigg\} \\
&= \mathbb{E}_{\mathcal{N}(\boldsymbol{\eta}\,;\mathbf{0},\mathbf{I})}\left[\nabla_{\boldsymbol{\mu}}\left\{\log p\big(\mathbf{X},T^{-1}(S^{-1}(\boldsymbol{\eta}))\big) + \log\big|\det J_{T^{-1}}\big(S^{-1}(\boldsymbol{\eta})\big)\big|\right\}\right] \\
&= \mathbb{E}_{\mathcal{N}(\boldsymbol{\eta}\,;\mathbf{0},\mathbf{I})}\left[\nabla_{\boldsymbol{\theta}}\log p\big(\mathbf{X},\boldsymbol{\theta}\big)\nabla_{\boldsymbol{\zeta}}T^{-1}(\boldsymbol{\zeta})\nabla_{\boldsymbol{\mu}}S_{\boldsymbol{\mu},\boldsymbol{\omega}}^{-1}(\boldsymbol{\eta}) + \nabla_{\boldsymbol{\zeta}}\log\big|\det J_{T^{-1}}(\boldsymbol{\zeta})\big|\nabla_{\boldsymbol{\mu}}S_{\boldsymbol{\mu},\boldsymbol{\omega}}^{-1}(\boldsymbol{\eta})\right] \\
&= \mathbb{E}_{\mathcal{N}(\boldsymbol{\eta}\,;\mathbf{0},\mathbf{I})}\left[\nabla_{\boldsymbol{\theta}}\log p\big(\mathbf{X},\boldsymbol{\theta}\big)\nabla_{\boldsymbol{\zeta}}T^{-1}(\boldsymbol{\zeta}) + \nabla_{\boldsymbol{\zeta}}\log\big|\det J_{T^{-1}}(\boldsymbol{\zeta})\big|\right]
\end{aligned}
$$

Similarly, consider the gradient with respect to the $\boldsymbol{\omega}$ parameter of the standardization. The gradient with respect to a single component, $\omega_k$, has a clean form. We abuse the $\nabla$ notation to maintain consistency with the rest of the text (instead of switching to $\partial$).

$$
\begin{aligned}
\nabla_{\omega_k}\mathcal{L} &= \nabla_{\omega_k}\Bigg\{\mathbb{E}_{\mathcal{N}(\boldsymbol{\eta}\,;\mathbf{0},\mathbf{I})}\left[\log p\big(\mathbf{X},T^{-1}(S_{\boldsymbol{\mu},\boldsymbol{\omega}}^{-1}(\boldsymbol{\eta}))\big) + \log\big|\det J_{T^{-1}}\big(S_{\boldsymbol{\mu},\boldsymbol{\omega}}^{-1}(\boldsymbol{\eta})\big)\big|\right] \\
&\qquad + \frac{K}{2}(1+\log(2\pi)) + \sum_{k=1}^{K}\log(\exp(\omega_k))\Bigg\} \\
&= \mathbb{E}_{\mathcal{N}(\eta_k)}\left[\nabla_{\omega_k}\left\{\log p\big(\mathbf{X},T^{-1}(S_{\boldsymbol{\mu},\boldsymbol{\omega}}^{-1}(\boldsymbol{\eta}))\big) + \log\big|\det J_{T^{-1}}\big(S_{\boldsymbol{\mu},\boldsymbol{\omega}}^{-1}(\boldsymbol{\eta})\big)\big|\right\}\right] + 1 \\
&= \mathbb{E}_{\mathcal{N}(\eta_k)}\left[\big(\nabla_{\theta_k}\log p\big(\mathbf{X},\boldsymbol{\theta}\big)\nabla_{\zeta_k}T^{-1}(\boldsymbol{\zeta}) + \nabla_{\zeta_k}\log\big|\det J_{T^{-1}}(\boldsymbol{\zeta})\big|\big)\nabla_{\omega_k}S_{\boldsymbol{\mu},\boldsymbol{\omega}}^{-1}(\boldsymbol{\eta}))\right] + 1. \\
&= \mathbb{E}_{\mathcal{N}(\eta_k)}\left[\big(\nabla_{\theta_k}\log p\big(\mathbf{X},\boldsymbol{\theta}\big)\nabla_{\zeta_k}T^{-1}(\boldsymbol{\zeta}) + \nabla_{\zeta_k}\log\big|\det J_{T^{-1}}(\boldsymbol{\zeta})\big|\big)\eta_k\exp(\omega_k)\right] + 1.
\end{aligned}
$$

## C Running ADVI in Stan

Visit `http://mc-stan.org/` to download the latest version of Stan. Follow instructions on how to install Stan. You are then ready to use ADVI.

Stan offers multiple interfaces. We describe the command line interface (`cmdStan`) below.

The syntax is

```
./myModel    variational
             grad_samples=M              ( M = 1 default )
             data file=myData.data.R
             output file=output_advi.csv
             diagnostic_file=elbo_advi.csv
```

where `myData.data.R` is the dataset stored in the R language `Rdump` format. `output_advi.csv` contains samples from the posterior and `elbo_advi.csv` reports the ELBO.

## D Transformations of Continuous Probability Densities

We present a brief summary of transformations, largely based on [2].

Consider a univariate (scalar) random variable $X$ with probability density function $f_X(x)$. Let $\mathcal{X} = \operatorname{supp}(f_X(x))$ be the support of $X$. Now consider another random variable $Y$ defined as $Y = T(X)$. Let $\mathcal{Y} = \operatorname{supp}(f_Y(y))$ be the support of $Y$.

If $T$ is a one-to-one and differentiable function from $\mathcal{X}$ to $\mathcal{Y}$, then $Y$ has probability density function

$$ f_Y(y) = f_X\left(T^{-1}(y)\right)\left|\frac{\mathrm{d}T^{-1}(y)}{\mathrm{d}y}\right|. $$

Let us sketch a proof. Consider the cumulative density function $Y$. If the transformation $T$ is increasing, we directly apply its inverse to the cdf of $Y$. If the transformation $T$ is decreasing, we apply its inverse to one minus the cdf of $Y$. The probability density function is the derivative of the cumulative density function. These things combined give the absolute value of the derivative above.

The extension to multivariate variables $X$ and $Y$ requires a multivariate version of the absolute value of the derivative of the inverse transformation. This is the the absolute determinant of the Jacobian, $|\det J_{T^{-1}}(Y)|$ where the Jacobian is

$$ J_{T^{-1}}(Y) = \begin{pmatrix} \frac{\partial T_1^{-1}}{\partial y_1} & \cdots & \frac{\partial T_1^{-1}}{\partial y_K} \\ \vdots & & \vdots \\ \frac{\partial T_K^{-1}}{\partial y_1} & \cdots & \frac{\partial T_K^{-1}}{\partial y_K} \end{pmatrix}. $$

Intuitively, the Jacobian describes how a transformation warps unit volumes across spaces. This matters for transformations of random variables, since probability density functions must always integrate to one.

## E Setting a Stepsize Sequence for ADVI

We use adaGrad [3] to adaptively set the stepsize sequence in ADVI. While adaGrad offers attractive convergence properties, it can be slow for non-convex problems. One reason is because it has infinite memory. (It tracks the norm of the gradient starting from the beginning of the optimization.) In ADVI we randomly initialize the variational approximation, which can be far from the true posterior. This makes adaGrad take very small steps for the rest of the optimization, thus slowing convergence. Limiting adaGrad's memory speeds up convergence in practice, an effect also observed in training neural networks [4]. (See [5] for an analysis of these trade-offs and a method that combines benefits from both.)

Consider the stepsize $\boldsymbol{\rho}^{(i)}$ and a gradient vector $\boldsymbol{g}^{(i)}$ at iteration $i$. In adaGrad, $k$th element of $\boldsymbol{\rho}^{(i)}$ is

$$\rho_k^{(i)} = \frac{\eta}{\tau + \sqrt{s_k^{(i)}}}.$$

The vector $\boldsymbol{s}$ is the gradient vector squared element-wise and summed over all times steps since the start of the optimization. Instead, we limit this by recursively downweighting previous iterations as

$$s_k^{(i)} = 0.9 \times s_k^{(i-1)} + 0.1 \times g_k^{2(i)}.$$

We do a grid search for the scaling coefficient $\eta$ and, following Hoffman et al. [6], set the offset $\tau = 1$.

## F  Linear Regression with Automatic Relevance Determination

Linear regression with automatic relevance determination (ARD) is a high-dimensional sparse regression model [7, 8]. We describe the model below. Stan code is in Figure 6.

The inputs are $\mathbf{X} = \boldsymbol{x}_{1:N}$ where each $\boldsymbol{x}_n$ is $D$-dimensional. The outputs are $\boldsymbol{y} = y_{1:N}$ where each $y_n$ is 1-dimensional. The weights vector $\boldsymbol{w}$ is $D$-dimensional. The likelihood

$$p(\boldsymbol{y} \mid \mathbf{X}, \boldsymbol{w}, \sigma) = \prod_{n=1}^N \mathcal{N}\left(y_n \mid \boldsymbol{w}^\top \boldsymbol{x}_n , \sigma\right)$$

describes measurements corrupted by iid Gaussian noise with unknown standard deviation $\sigma$.

The ARD prior and hyper-prior structure is as follows

$$p(\boldsymbol{w}, \sigma, \boldsymbol{\alpha}) = p(\boldsymbol{w}, \sigma \mid \boldsymbol{\alpha}) p(\boldsymbol{\alpha})$$

$$= \mathcal{N}\left(\boldsymbol{w} \mid 0 , \sigma\left(\mathrm{diag}\sqrt{\boldsymbol{\alpha}}\right)^{-1}\right) \mathrm{InvGam}(\sigma \mid a_0, b_0) \prod_{i=1}^D \mathrm{Gam}(\alpha_i \mid c_0, d_0)$$

where $\boldsymbol{\alpha}$ is a $D$-dimensional hyper-prior on the weights, where each component gets its own independent Gamma prior.

We simulate data such that only half the regressions have predictive power. The results in Figure 4a use $a_0 = b_0 = c_0 = d_0 = 1$ as hyper-parameters for the Gamma priors.

## G  Hierarchical Logistic Regression

Hierarchical logistic regression models structured datasets in an intuitive way. We study a model of voting preferences from the 1988 United States presidential election. Chapter 14.1 of [9] motivates the model and explains the dataset. We also describe the model below. Stan code is in Figure 7, based on [10].

$$\Pr(y_n = 1) = \mathrm{sigmoid}\Bigg(\beta^0 + \beta^{\mathrm{female}} \cdot \mathrm{female}_n + \beta^{\mathrm{black}} \cdot \mathrm{black}_n + \beta^{\mathrm{female.black}} \cdot \mathrm{female.black}_n$$

$$+ \alpha_{k[n]}^{\mathrm{age}} + \alpha_{l[n]}^{\mathrm{edu}} + \alpha_{k[n],l[n]}^{\mathrm{age.edu}} + \alpha_{j[n]}^{\mathrm{state}}\Bigg)$$

$$\alpha_j^{\mathrm{state}} \sim \mathcal{N}\left(\alpha_{m[j]}^{\mathrm{region}} + \beta^{\mathrm{v.prev}} \cdot \mathrm{v.prev}_j , \sigma_{\mathrm{state}}\right).$$

The hierarchical variables are

$$\alpha_k^{\mathrm{age}} \sim \mathcal{N}\left(0 , \sigma_{\mathrm{age}}\right) \text{ for } k = 1, \dots, K$$
$$\alpha_l^{\mathrm{edu}} \sim \mathcal{N}\left(0 , \sigma_{\mathrm{edu}}\right) \text{ for } l = 1, \dots, L$$
$$\alpha_{k,l}^{\mathrm{age.edu}} \sim \mathcal{N}\left(0 , \sigma_{\mathrm{age.edu}}\right) \text{ for } k = 1, \dots, K, l = 1, \dots, L$$
$$\alpha_m^{\mathrm{region}} \sim \mathcal{N}\left(0 , \sigma_{\mathrm{region}}\right) \text{ for } m = 1, \dots, M.$$

The standard deviation terms all have uniform hyper-priors, constrained between 0 and 100.

## H  Non-negative Matrix Factorization: Constrained Gamma Poisson Model

The Gamma Poisson factorization model describes discrete data matrices [11, 12].

Consider a $U \times I$ matrix of observations. We find it helpful to think of $u = \{1, \cdots, U\}$ as users and $i = \{1, \cdots, I\}$ as items, as in a recommendation system setting. The generative process for a Gamma Poisson model with $K$ factors is

1. For each user $u$ in $\{1, \cdots, U\}$:
   - For each component $k$, draw $\theta_{uk} \sim \text{Gam}(a_0, b_0)$.
2. For each item $i$ in $\{1, \cdots, I\}$:
   - For each component $k$, draw $\beta_{ik} \sim \text{Gam}(c_0, d_0)$.
3. For each user and item:
   - Draw the observation $y_{ui} \sim \text{Poisson}(\boldsymbol{\theta}_u^\top \boldsymbol{\beta}_i)$.

A potential downfall of this model is that it is not uniquely identifiable: swapping rows and columns of $\boldsymbol{\theta}$ and $\boldsymbol{\beta}$ give the same inner product. One way to contend with this is to constrain either vector to be an ordered vector during inference. We constrain each $\boldsymbol{\theta}_u$ vector in our model in this fashion. Stan code is in Figure 8. We set $K = 10$ and all the Gamma hyper-parameters to 1 in our experiments.

## I  Non-negative Matrix Factorization: Dirichlet Exponential Model

Another model for discrete data is a Dirichlet Exponential model. The Dirichlet enforces uniqueness while the exponential promotes sparsity. This is a non-conjugate model that does not appear to have been studied in the literature.

The generative process for a Dirichlet Exponential model with $K$ factors is

1. For each user $u$ in $\{1, \cdots, U\}$:
   - Draw the $K$-vector $\boldsymbol{\theta}_u \sim \text{Dir}(\boldsymbol{\alpha}_0)$.
2. For each item $i$ in $\{1, \cdots, I\}$:
   - For each component $k$, draw $\beta_{ik} \sim \text{Exponential}(\lambda_0)$.
3. For each user and item:
   - Draw the observation $y_{ui} \sim \text{Poisson}(\boldsymbol{\theta}_u^\top \boldsymbol{\beta}_i)$.

Stan code is in Figure 9. We set $K = 10$, $\alpha_0 = 1000$ for each component, and $\lambda_0 = 0.1$. With this configuration of hyper-parameters, the factors $\boldsymbol{\beta}_i$ appear sparse.

## J  Gaussian Mixture Model

The Gaussian mixture model (GMM) is a celebrated probability model. We use it to group a dataset of natural images based on their color histograms. We build a high-dimensional GMM with a Gaussian prior for the mixture means, a lognormal prior for the mixture standard deviations, and a Dirichlet prior for the mixture components.

The images are in $\mathbf{Y} = \boldsymbol{y}_{1:N}$ where each $\boldsymbol{y}_n$ is $D$-dimensional and there are $N$ observations. The likelihood for the images is

$$p(\mathbf{Y} \mid \boldsymbol{\theta}, \boldsymbol{\mu}, \boldsymbol{\sigma}) = \prod_{n=1}^{N} \sum_{k=1}^{K} \theta_k \prod_{d=1}^{D} \mathcal{N}(y_{nd} \mid \mu_{kd}, \sigma_{kd})$$

with a Dirichlet prior for the mixture proportions

$$p(\boldsymbol{\theta}) = \text{Dir}(\boldsymbol{\theta} ; \boldsymbol{\alpha}_0),$$

a Gaussian prior for the mixture means

$$p(\boldsymbol{\mu}) = \prod_{k=1}^{D} \prod_{d=1}^{D} \mathcal{N}(\mu_{kd}\,;\,0,1)$$

and a lognormal prior for the mixture standard deviations

$$p(\boldsymbol{\sigma}) = \prod_{k=1}^{D} \prod_{d=1}^{D} \mathrm{logNormal}(\sigma_{kd}\,;\,0,1)$$

The dimension of the color histograms in the imageCLEF dataset is $D = 576$. This is a concatenation of three 192-length histograms, one for each color channel (red, green, blue) of the images.

We scale the image histograms to have zero mean and unit variance. Setting $\alpha_0$ to a small value encourages the model to use fewer components to explain the data. Larger values of $\alpha_0$ encourage the model to use all $K$ components. We set $\alpha_0 = 1\,000$ in our experiments.

ADVI code is in Figure 10. The stochastic data subsampling version of the code is in Figure 11.

```
data {
  int<lower=0> N;    // number of data items
  int<lower=0> D;    // dimension of input features
  matrix[N,D]  x;    // input matrix
  vector[N]    y;    // output vector

  // hyperparameters for Gamma priors
  real<lower=0> a0;
  real<lower=0> b0;
  real<lower=0> c0;
  real<lower=0> d0;
}

parameters {
  vector[D] w;                // weights (coefficients) vector
  real<lower=0> sigma;        // standard deviation
  vector<lower=0>[D] alpha;   // hierarchical latent variables
}

transformed parameters {
  vector[D] one_over_sqrt_alpha;
  for (i in 1:D) {
    one_over_sqrt_alpha[i] <- 1 / sqrt(alpha[i]);
  }
}

model {
  // alpha: hyper-prior on weights
  alpha ~ gamma(c0, d0);

  // sigma: prior on standard deviation
  sigma ~ inv_gamma(a0, b0);

  // w: prior on weights
  w ~ normal(0, sigma * one_over_sqrt_alpha);

  // y: likelihood
  y ~ normal(x * w, sigma);
}
```

**Figure 6:** Stan code for Linear Regression with Automatic Relevance Determination.

```
data {
  int<lower=0> N;
  int<lower=0> n_age;
  int<lower=0> n_age_edu;
  int<lower=0> n_edu;
  int<lower=0> n_region_full;
  int<lower=0> n_state;
  int<lower=0,upper=n_age> age[N];
  int<lower=0,upper=n_age_edu> age_edu[N];
  vector<lower=0,upper=1>[N] black;
  int<lower=0,upper=n_edu> edu[N];
  vector<lower=0,upper=1>[N] female;
  int<lower=0,upper=n_region_full> region_full[N];
  int<lower=0,upper=n_state> state[N];
  vector[N] v_prev_full;
  int<lower=0,upper=1> y[N];
}
parameters {
  vector[n_age] a;
  vector[n_edu] b;
  vector[n_age_edu] c;
  vector[n_state] d;
  vector[n_region_full] e;
  vector[5] beta;
  real<lower=0,upper=100> sigma_a;
  real<lower=0,upper=100> sigma_b;
  real<lower=0,upper=100> sigma_c;
  real<lower=0,upper=100> sigma_d;
  real<lower=0,upper=100> sigma_e;
}
transformed parameters {
  vector[N] y_hat;

  for (i in 1:N)
    y_hat[i] <- beta[1]
                + beta[2] * black[i]
                + beta[3] * female[i]
                + beta[5] * female[i] * black[i]
                + beta[4] * v_prev_full[i]
                + a[age[i]]
                + b[edu[i]]
                + c[age_edu[i]]
                + d[state[i]]
                + e[region_full[i]];
}
model {
  a ~ normal (0, sigma_a);
  b ~ normal (0, sigma_b);
  c ~ normal (0, sigma_c);
  d ~ normal (0, sigma_d);
  e ~ normal (0, sigma_e);
  beta ~ normal(0, 100);
  y ~ bernoulli_logit(y_hat);
}
```

**Figure 7:** Stan code for Hierarchical Logistic Regression, from [10].

```
data {
  int<lower=0> U;
  int<lower=0> I;
  int<lower=0> K;
  int<lower=0> y[U,I];
  real<lower=0> a;
  real<lower=0> b;
  real<lower=0> c;
  real<lower=0> d;
}

parameters {
  positive_ordered[K] theta[U]; // user preference
  vector<lower=0>[K] beta[I];    // item attributes
}

model {
  for (u in 1:U)
    theta[u] ~ gamma(a, b); // componentwise gamma
  for (i in 1:I)
    beta[i] ~ gamma(c, d);  // componentwise gamma

  for (u in 1:U) {
    for (i in 1:I) {
      y[u,i] ~ poisson(theta[u]'*beta[i]);
    }
  }
}
```

**Figure 8:** Stan code for Gamma Poisson non-negative matrix factorization model.

```
data {
  int<lower=0> U;
  int<lower=0> I;
  int<lower=0> K;
  int<lower=0> y[U,I];
  real<lower=0> lambda0;
  real<lower=0> alpha0;
}

transformed data {
  vector<lower=0>[K] alpha0_vec;
  for (k in 1:K) {
    alpha0_vec[k] <- alpha0;
  }
}

parameters {
  simplex[K] theta[U];          // user preference
  vector<lower=0>[K] beta[I];   // item attributes
}

model {
  for (u in 1:U)
    theta[u] ~ dirichlet(alpha0_vec); // componentwise dirichlet
  for (i in 1:I)
    beta[i] ~ exponential(lambda0);   // componentwise exponential

  for (u in 1:U) {
    for (i in 1:I) {
      y[u,i] ~ poisson(theta[u]'*beta[i]);
    }
  }
}
```

**Figure 9:** Stan code for Dirichlet Exponential non-negative matrix factorization model.

```
data {
  int<lower=0> N;          // number of data points in entire dataset
  int<lower=0> K;          // number of mixture components
  int<lower=0> D;          // dimension
  vector[D] y[N];          // observations

  real<lower=0> alpha0; // dirichlet prior
}

transformed data {
  vector<lower=0>[K] alpha0_vec;
  for (k in 1:K)
    alpha0_vec[k] <- alpha0;
}

parameters {
  simplex[K] theta;                   // mixing proportions
  vector[D] mu[K];                    // locations of mixture components
  vector<lower=0>[D] sigma[K];        // standard deviations of mixture components
}

model {
  // priors
  theta ~ dirichlet(alpha0_vec);
  for (k in 1:K) {
      mu[k] ~ normal(0.0, 1.0);
      sigma[k] ~ lognormal(0.0, 1.0);
  }

  // likelihood
  for (n in 1:N) {
    real ps[K];
    for (k in 1:K) {
      ps[k] <- log(theta[k]) + normal_log(y[n], mu[k], sigma[k]);
    }
    increment_log_prob(log_sum_exp(ps));
  }
}
```

**Figure 10:** Stan code for the GMM example.

```
functions {
  real divide_promote_real(int x, int y) {
    real x_real;
    x_real <- x;
    return x_real / y;
  }
}

data {
  int<lower=0> NFULL;      // total number of datapoints in dataset
  int<lower=0> N;          // number of data points in minibatch

  int<lower=0> K;          // number of mixture components
  int<lower=0> D;          // dimension

  vector[D] yFULL[NFULL];  // dataset
  vector[D] y[N];          // minibatch

  real<lower=0> alpha0;    // dirichlet hyper-prior parameter
}

transformed data {
  real minibatch_factor;
  vector<lower=0>[K] alpha0_vec;
  for (k in 1:K) {
    alpha0_vec[k] <- alpha0 / K;
  }
  minibatch_factor <- divide_promote_real(N, NFULL);
}

parameters {
  simplex[K] theta;                        // mixing proportions
  vector[D] mu[K];                         // locations of mixture components
  vector<lower=0>[D] sigma[K];  // standard deviations of mixture components
}

model {
  // priors
  theta ~ dirichlet(alpha0_vec);
  for (k in 1:K) {
      mu[k] ~ normal(0.0, 1.0);
      sigma[k] ~ lognormal(0.0, 1.0);
  }

  // likelihood
  for (n in 1:N) {
    real ps[K];
    for (k in 1:K) {
      ps[k] <- log(theta[k]) + normal_log(y[n], mu[k], sigma[k]);
    }
    increment_log_prob(log_sum_exp(ps));
  }
  increment_log_prob(log(minibatch_factor));
}
```

**Figure 11:** Stan code for the GMM example, with stochastic subsampling of the dataset.