[Reviews · NeurIPS 2015]

Submitted by Assigned_Reviewer_1

*** This is a light review! ***

The paper combines several contemporary aspects of variational inference to make an automatic variational inference algorithm using the STAN framework. Models are written in the STAN language, and then a factorized Gaussian approximation is used in STAN's continuous space (presumably the same space that is used for MCMC?)

This is a really good contribution to the field. It makes VI applicable to a really wide range of models (anywhere where HMC can be run, I presume?).

It's a shame that there's no way to correlate the posterior, but I guess that would make for tidy future work.

Despite enjoying the paper, I have reservations about whether NIPS is really the right venue, hence my relatively low score.

There's little technical contribution here but there is a huge amount of engineering effort which makes good use of recently proposed methods.

Without the software availability, this paper would be of limited value, but the software itself is undoubtedly of interest to many in the community.
Summary: A really excellent bit of work, making variational inference available in STAN. Somewhat lacking technical contribution, but of interest to many in the community nonetheless.

Submitted by Assigned_Reviewer_2

For improvement, I would like to suggest adding the following:

(1) A table that presents possible models that the code package can handle. (2) A schematic for structure of ADVI code: Data, parameters, transformed parameters, model. For each part, what are the essential quantities that the user to specify? I don't think it should be as detailed as a user manual, but at least it should be informative enough for a reader of the paper to have a rough idea of how one could use the package (rather than assuming the reader to look into their supplementary material).

Other suggestions: (1) Examples in 3.1: the posteriors under those models aren't Gaussian, but still they are single mode (per datapoint) and their density is easily captured by a Gaussian (in transformed space). Can authors show other more complicated examples, such as multi-modal posteriors?

(2) Comparison: I am not quite sure why the authors compare ADVI to sampling based methods. Can authors compare ADVI to other model-specific code for VI (the same inference method) and still say ADVI is fast and accurate (if not faster or more accurate, as much as the code packages written for the specific models)? This will be much more convincing for showing its usefulness.

=============== After reading their rebuttal

================

Thanks for answering my questions. Of course, the customised version would be faster/more efficient/(maybe) more accurate, but I would want to know how it compares with ADVI for those non-conjugate cases. I will keep my rating the same. Thanks!
Summary: The paper presents a code package written in Stan for variational inference under a variety of models that are widely used in machine learning literature. I believe automatising (relatively) standard inference methods like variational Bayes should be more encouraged, so that scientists in this community can spend most of their time doing more creative work, rather than deriving equations and writing code for the "standard" inference methods like VB, for testing their data/models. While I believe this line of research could make a significant contribution to the community, this paper itself is less informative than it should (or could) be due to the lack of structure in the paper.

Submitted by Assigned_Reviewer_3

Review Summary ============== Score: 8 top 50% of NIPS papers

I'm excited about the core ideas in this paper: it's a huge step towards making variational inference general-purpose at scale for a big class of models. I think it belongs at NIPS, especially if the authors can address some concerns about local optima and revise the paper to include a new plot with a more intuitive illustrative example.

Summary of Paper ================ This paper develops a method for automatic variational inference in the Stan probabilistic programming language, which can support any model that has a differentiable log joint probability. The proposed method works by transforming all latent variables to unconstrained real values, and then approximating their posterior with a Gaussian with diagonal covariance (the diagonal assumption corresponds to an mean-field independence assumption on every scalar in the parameter vector). A further transformation standardizes each 1D Gaussian to a standard normal (zero mean, unit variance), so that Monte Carlo integration can be used to estimate the expectations involved in the variational objective function. To scale to massive datasets, the authors employ stochastic variational inference with an adaptive learning rate. Experiments show that the method provides better predictions on heldout data than Stan's MCMC algorithm for a variety of nonconjugate models (regression, matrix factorization, and mixtures).

Reviewer Comments on Method =========================== Overall, I like the general approach used. I have a few key issues I'd like addressed in the rebuttal, in order of importance.

Issue M1: I'd like to see at least some discussion of how vulnerable this method is to local optima. My guess is, that like other variational methods, the answer is extremely sensitive. It would be interesting to hear if the authors have any strategies to avoid this sensitivity, aside from multiple random initializations.

Issue M2: I'm concerned that the monte-carlo integration strategy may be very noisy and lead to slow convergence. Previous work on black-box variational (Ranganath, Gerish and Blei 2014) indicated that without some strategies to reduce variance (rao-blackwell-ization and control variate methods), simple monte carlo estimates to obtain gradients were quite terrible -- see esp. their Fig. 2. I'd like to know if this kind of diagnostic plot, or any other, was ever made. The experiments seem to converge OK, but I find it really surprising that a single sample (M=1), as used in all experiments, is sufficient.

Issue M3: I'd like to see more discussion of how the method is initialized. This is a crucial for variational inference methods that get stuck easily in local optima. Have different strategies been tried?

Issue M4: How to select batch size? It seems like the growing consensus is that very small batches will lead to poor models, which makes sense. But is there any advice to practioners about how to select an appropriate batch size? Perhaps just use the largest size you can afford, doing full-dataset inference if feasible?

Reviewer Comments on Experiments ================================ Issue E1: Internal experiments should show many (at least 5, preferablly more like 20 or 50) runs of each method (each from an independent initialization), rather than just one run. Local optima are very prevalent in inference, and plots of multiple runs can help diagnose how sensitive each method (MCMC/ADVI) is.

Issue E2: Experiments are missing one "intuition-building" plot, that shows the estimated parameters in a case for which the good or ideal parameters are "obvious" from inspection of the raw data. I like the faces in Fig 5c, but its hard to tell if this is a "good" result. Instead, I recommend something like a classic mixture model on 2D scatterplot data with a few obvious clusters, some of which have non-trivial covariance. The result from ADVI should be some Gaussian blobs centered on cluster means but with diagonal covariances (due to mean-field), while the MCMC result maybe be more accurate (no independence assumptions) but probably slower. I think this will help readers understand the tradeoffs of mean-field, etc. It's often really hard to understand from traceplots whether a given run is "good" in an absolute sense.

Issue E3: In Line 414, "label switching" is blamed for the poor performance of MCMC for the mixture model. I don't think I buy this explanation. The fact that cluster indices can be permutted over many iterations doesn't explain poor predictive performance, because prediction scores should be irrelevant to the order of the clusters. Instead, the problem is probably that the MCMC sampler is stuck in a local optima and cannot escape. This is the kind of thing that is best diagnosed with the plot from E2.

Less Critical Issues (need not reply to in rebuttal) ---------------------------------------------------- Issue M5: Variational and MCMC are often seen as either/or choices, but I'm often interested in whether variational could be used to rapidly get to a decent estimate, and then MCMC is used to get a more refined, reliable estimate. Is it possible to initialize an MCMC chain in Stan with the output of variational? Or vice versa?

Issue E4: This may be too much to ask, but I think it would be nice to include one comparison to an external method (variational probably, but MCMC could work too) that has been custom-designed for some specific data and model. For many readers, the question of whether they should use Stan or roll-their-own custom solution may be relevant, and it would be interesting to show a case for which the Stan approach reaches comparable performance without the cost of custom inference.

Significance ============ This project has great potential for wide-adoption, given its complete integration into the increasingly-popular Stan language.

Novelty ======= Developing the proposed general-purpose framework and integrating it into Stan is definitely a very new and worthwhile contribution.

Clarity ======= The paper has a crisp, simple style that I found very accessible. Aside from a few unclear sentences (detailed below), I think the overall organization and style do not need heavy revision.

Line-by-line Comments ===================== (not necessary to respond to these in rebuttal)

Line 257: I found the phrase "we cannot directly represent it as a computer program" a bit confusing. Perhaps just leave it out. The sentence about an intractable integral is enough.

Supplement E, Line 589: You give ref 3 (the SVI JMLR article) as a reference for adagrad, but in fact that paper has very little material on adaptive gradients, aside from a reference other papers. Please fix.
Summary: I'm excited about the core ideas in this paper: it's a huge step towards making variational inference general-purpose at scale for a big class of models. I think it belongs at NIPS, especially if the authors can address some concerns about local optima and revise the paper to include a new plot with a more intuitive illustrative example.

Submitted by Assigned_Reviewer_4

Variational inference has proven a useful tool for distribution approximation, in particular due to its speed relative to, e.g., Monte Carlo methods. However, as it typically involves choosing an appropriate approximating distribution form and tailoring calculations for that form, it can be difficult for practitioners to apply. The current work addresses this problem by transforming the variables of the distribution such that their support is some full Euclidean space. Then the authors apply a Gaussian variational approximation in this transformed space. The authors also say that they provide code within the Stan package that allows practitioners to input a Bayesian generative model, and their method works automatically from there.

This paper addresses an important problem: automating posterior approximation in a user-friendly and computationally efficient way. The major contributions seem to be the parameter-transformation together with Gaussian approximation in the transformed space, the demonstration of the effectiveness of this method, and the provision of usable code as part of Stan. The authors clearly describe relevant previous work and their own methodology. This paper seems like a strong contribution to the field.

Some additional comments/questions appear below.

Major comments

The choice of "Gaussian variational distribution" (e.g. p. 3, line 099) in the transformed parameter space is an interesting choice. It would be nice to see this choice explored in more detail. Is there any intuition (or theoretical or empirical evidence) about when this would be a bad choice over alternative distributions? Or why should we expect this to be a good choice?

On p. 3, line 137--138, we learn that we must use continuous variables. While the authors argue that many models can be cast as "differentiable probability models," one wonders what the limitations of this are. What if we are interested in the indicator variables? Can they be obtained after calculating the posterior with discrete variables marginalized out? What models *don't* work here?

Minor comments

Fig 1: How is the average log predictive calculated for ADVI vs NUTS? It's often the case that approximations are made in calculating the log predictive; it would be good to know if that were the case here and whether different approximations went into calculating the log predictive for ADVI vs NUTS.

p. 2, line 073: "ADVI is orders of magnitude faster than NUTS" Can we get some numbers or a reference to a figure, table, etc to support this? Does this mean just in terms of running time on a fixed data-set-size or in terms of reaching a desired "average log predictive"?

--- line 075: "[on other models,] we consistently observe speed-up against NUTS" again. I understand the authors are giving a sneak peak, but it would be nice to see some references to numbers (e.g. see Sec X for exact numbers), if not numbers right here.

p. 2, line 076: "size of data we more commonly find in machine learning"

I think this statement is unfounded and vague. It's certainly a larger data size and limiting for slow algorithms. But it seems a bit much to say that it's the most common actual data set size in the practice of machine learning (where "common" is surely ill-defined to start with). We might just have interest in these sizes because they're a bottleneck for algorithms, not because they're common.

p.2, line 086: "build on and extend these ideas" This statement is pretty vague. Perhaps the authors could give a high-level sense of what the contribution of the new work here is going to be.

p. 4, line 168: "In classical variational inference, we typically design a conditionally conjugate model; the optimal approximating family matches the prior, which satisfies the support constraint by definition [16]"

--- (1) This is true for exponential families, which I think haven't yet been mentioned by this point, although implicitly all of the conjugacy seems to refer to exponential family conjugacy. It might be a good idea to make the exponential family assumptions explicit before this point. This particular statement is true for exponential families presumably entirely because the support can't depend on the parameter by construction. But for something like a Pareto, it seems this wouldn't be true.

--- (2) Bishop (ref [16]) is a huge book. If referencing Bishop, make sure to provide a page number.

p. 4, line 193: "warps unit volumes [17]" Please add page numbers when citing books. I don't strictly think you need this citation (if you find yourself low on citation space), but it's always nice to provide references.

--- see also book citations [19] and [20]: please add a chapter reference or page numbers

Fig 3: Very nice figure for illustration of the method.

p. 4, line 202: "Now, we can choose a variational distribution independent from the model." Are there any side effects of this? Is this ever a bad idea? E.g., does it ever give a bad variational approximation?

p. 6, line 274: I guess the initializations must not matter too much, but I wonder if something better than 0 could be used. Or if there's some reason (e.g. pre-processing of the data to have empirical mean and variance 0 and 1, respectively) to think that mu = 0 and omega = 0 would be reasonable.

p. 6, line 312: "(typically between 1 and 10)" I'm surprised how low this is. How is this enough? (I agree it's supported by the empirical evidence in Fig 4, but I'd like to understand why.)

--- also line 363: "A single sample per iteration is sufficient" is this sufficiency determined empirically or are there theoretical reasons to think this should be true?

p. 6, line 320: In non-NUTS HMC, how were the step size and number of steps parameters set?

p. 7, line 339: "predictive accuracy" or predictive probability?

p. 7, line 340: "For the MCMC techniques, we plug in posterior samples into the likelihood" Why is this a good idea?

p. 7, line 342: "We initialize with a draw from a standard Gaussian." Why is this initialization needed? Why can't one draw directly from the posterior approximation?

Typos

p. 6, line 294: "independent term" should perhaps be "the independent term" or "independent terms"

-------------------------------------- Additional comments after author feedback --------------------------------------

Most of my questions/concerns were answered. I remain enthuasiastic about the paper and look forward to reading the final version.
Summary: The authors provide a novel method and code that approximates a Bayesian posterior given only a Bayesian generative model. While it makes a specific approximation (transforms the parameters and fits a Gaussian in the transformed space) and only works on models with continuous variables with certain supports, it seems to be a useful step in a pressing area: automating Bayesian inference.

Author Feedback
Author rebuttal: We thank the reviewers for their insightful comments. We are happy that all reviewers appreciated the impact of automatic variational inference. We are excited to report that ADVI was officially released in Stan 2.7 two weeks ago.

(R1) A table of models + schematic for ADVI

We have these, but removed both due to space constraints. We will add them to the supplement.

(R1) No multi-modal posterior examples

The examples in 3.2 and 3.3 have multi-modal posteriors.

(R1) Why not compare to custom variational inference code?

All the models we test are non-conjugate, for which there is no standard variational inference technique. Comparing to sampling is a stronger statement, since sampling is asymptotically exact. As for speed, custom code should beat ADVI. Intuitively, a system engineered to solve a specific problem should outperform a system designed to solve many problems.

(R2) M1: local optima

Great point. ADVI is just as sensitive to local optima as any other variational inference procedure. Empirically, stochastic optimization seems to help, but random initializations remain the primary way of handling this.

(R2) M2: variance of gradient estimator

We use a different, but related, estimator than Ranganath14; using the gradient of the log joint exploits the smoothness of the log joint to reduce variance. We do have a similar plot, but omitted it because showing convergence with M=1 samples (the fewest possible) implies that our gradient has very low variance.

(R2) M3: how to initialize ADVI

We initialize with a standard Gaussian (line 342). The results seem robust to initialization, though this may not generalize to all models.

(R2) M4: how to select the batch size

We refer to the SVI paper [3] for this (line 419).

(R2) E1: multiple runs for experiments

Good point. We will modify at least one of the plots to show many random initializations.

(R2) E2: consistency plot

We agree. We omitted results from a similar study. We will present such results in the journal version of the paper [in preparation], which will address both predictive accuracy and consistency.

(R2) E3: label switching and HMC

In HMC, the label switching effect can sometimes be the cause of local optima. However, we agree with you: we'll run NUTS a few times and report what happens.

(R3) Why use a Gaussian?

We use a Gaussian as it is a natural first choice. We are studying extensions, such as a Gaussian with a full covariance matrix and a mean-field Student-t.

(R3) Discrete models

Purely discrete models, such as Ising models, restricted Boltzmann machines, and discrete neural networks, or untruncated Bayesian nonparametric models won't work. However, indicator variables, such as the assignment of a datapoint to a particular mixture can be computed after fitting the marginalized model [Murphy 2012, 11.2.3].

(R3) How are average log predictives calculated?

We approximate the predictive distributions in the same way for both ADVI and NUTS: we draw from the approximate posterior and plug into the likelihood to form MC estimates of the predictive.

(R3) How much faster than NUTS?

We will clarify these "sneak peeks" and provide numbers.

(R3) Vague statements about common dataset sizes and contributions.

Great points. We will clarify both issues in the text.

(R3) Conditionally conjugate models

You are right, we will ground this statement within the exponential family. We will also add page numbers to all book citations.

(R3) Independently choosing the variational approximation: a bad idea?

We decouple the choice from the model to help in automating variational inference. If the model exhibits exponential family conditional conjugacies, then the optimal mean-field variational family to use is defined by these conjugacy relationships [Bishop 2006, 10.2.1]. In general, the optimal variational family may not be normalizable. It is common in these settings to use a Gaussian, as in [Wang and Blei 2013, JMLR].

(R3) Do you recommend preprocessing the dataset?

Yes. Stan recommends the preprocessing routine you suggest, even when using NUTS.

(R3) Variance of the gradient

A great point that R2 also raises. Please see our answer above.

(R3) How were the parameters of HMC set?

Not carefully: just the defaults in Stan.

(R3) p7_340

See above for our answer to computing average log predictives.

(R4), (R5) and (R6) General comments

These are all great summaries. We gently point out that, as far as we know, the idea of using automated transformations of the model while fixing the variational distribution is a novel idea. It is understandable that this idea did not come across upon a light review of the paper. We agree that the immediate availability of ADVI in Stan is a strong factor in considering the value of this work.

(R6) Can I run ADVI wherever I can run HMC?

Absolutely!